# AICAr, a Widely Used AMPK Activator with Important AMPK-Independent Effects: A Systematic Review

**DOI:** 10.3390/cells10051095

**Published:** 2021-05-04

**Authors:** Dora Višnjić, Hrvoje Lalić, Vilma Dembitz, Barbara Tomić, Tomislav Smoljo

**Affiliations:** 1Laboratory of Cell Biology, Croatian Institute for Brain Research, University of Zagreb School of Medicine, 10000 Zagreb, Croatia; hrvoje.lalic@mef.hr (H.L.); vilma.dembitz@mef.hr (V.D.); barbara.tomic@mef.hr (B.T.); tomislav.smoljo@mef.hr (T.S.); 2Department of Physiology, University of Zagreb School of Medicine, 10000 Zagreb, Croatia

**Keywords:** AICAr, AMPK, metabolism, acadesine, exercise, cell cycle, purine, pyrimidine, cancer, leukemia

## Abstract

5-Aminoimidazole-4-carboxamide ribonucleoside (AICAr) has been one of the most commonly used pharmacological modulators of AMPK activity. The majority of early studies on the role of AMPK, both in the physiological regulation of metabolism and in cancer pathogenesis, were based solely on the use of AICAr as an AMPK-activator. Even with more complex models of AMPK downregulation and knockout being introduced, AICAr remained a regular starting point for many studies focusing on AMPK biology. However, there is an increasing number of studies showing that numerous AICAr effects, previously attributed to AMPK activation, are in fact AMPK-independent. This review aims to give an overview of the present knowledge on AMPK-dependent and AMPK-independent effects of AICAr on metabolism, hypoxia, exercise, nucleotide synthesis, and cancer, calling for caution in the interpretation of AICAr-based studies in the context of understanding AMPK signaling pathway.

## 1. Introduction

In 1995, 5-amino-4-imidazolecarboxamide (AICA) ribonucleoside or riboside was first proposed to be used as the activator of AMP-kinase (AMPK) in intact cells or, in other words, to play the same role that phorbol esters had in dissecting signaling pathways regulated by protein kinase C [1]. The study by Corton et al. was based on previous work showing that administration of the AICA riboside to intact cells causes AICA ribonucleotide or ribotide to accumulate inside the cell [2], and a study by Sullivan et al. [3] demonstrated that AICA ribotide mimicked the effect of AMP on allosteric activation of rat liver AMPK. Since then, the compound that has been widely used as an AMPK-agonist was an exogenous dephosphorylated AICA riboside that should be properly abbreviated AICAr. However, in more than 1700 articles that can be retrieved from PubMed on AMPK and AICA riboside, AICAr is often abbreviated as AICAR, although the AICAR acronym should be reserved for AICA ribotide or the phosphorylated form that is a physiological, endogenous precursor in de novo purine synthesis [4]. The nomenclature is additionally complicated because the other name used for the endogenous substance or AICAR is ZMP [5]. Moreover, the search of the literature reveals the common use of acadesine instead of AICAr [4,6,7,8,9,10,11,12,13,14,15,16,17,18,19].

As shown in Figure 1, AICAR or ZMP is an AMP analog. As a cell-permeable nucleotide, AICAr enters the cells through adenosine transporters [20] and becomes phosphorylated by adenosine kinase into AICAR [21]. AICAR or ZMP activates AMPK but it is 40- to 50- fold less potent than AMP in AMPK activation and accumulates in high concentrations in the cytoplasm [1], so that it was always likely that AICAr may have several AMPK-independent effects. Similar to AMP, AICAR binds to the γ subunit of AMPK, allosterically activates the enzyme, stimulates phosphorylation at Thr^172^ by liver kinase B1 (LKB1), and protects against pThr^172^ dephosphorylation [22,23]. Therefore, the most common method to test for AICAr-mediated activation of AMPK in particular tissues or cells is to detect the level of pThr^172^ AMPK by Western blot in lysates upon AICAr treatment. 

In this article, we will give a brief overview of the present knowledge on AMPK-dependent and AMPK-independent effects of AICAr.

## 2. AICAr, Metabolism and Diabetes

Since 1995, AICAr has been a valuable tool to identify metabolic pathways associated with AMPK activation in various tissues and cells in vivo and in vitro. In liver cells, AICAr was found to inhibit fatty acid and cholesterol synthesis [24], to increase fatty acid oxidation [25], and to inhibit gluconeogenesis by decreasing transcription of *PEPCK* [26]. In skeletal muscle cells, AICAr was found to activate glycogen phosphorylase and increase glycogenolysis [27], stimulate fatty acid oxidation and glucose uptake [28], and inhibit protein synthesis [29]. All of these studies supported the role of AMPK as a central metabolic hub of the cell that is activated whenever the ratio of AMP/ATP is high and which acts to increase catabolic and inhibit anabolic processes (Figure 2). 

However, investigators soon realized that ZMP in millimolar concentrations could exert many other effects, including modulation of other AMP-sensitive enzymes, like glycogen phosphorylase [30], and fructose-1-6-biphosphatase, a rate-controlling enzyme in gluconeogenesis, which is likely to explain why the effects of AICAr on gluconeogenesis in the liver are AMPK-independent [31]. The best way to assess the role of AMPK in the effects of AICAr in vivo could be provided by AMPK knockout mice. As shown in Table 1, data obtained by transgenic mice models revealed that AICAr-mediated effects on glucose uptake in skeletal muscle cells [32,33,34,35], lipogenesis and fatty acid oxidation in the liver [36], and decreased fat synthesis in adipose cells were AMPK-dependent [37]. However, a decrease in gluconeogenesis and an inhibition of oxidative phosphorylation (OXPHOS) can be observed in response to AICAr in mice lacking both AMPKα1 and α2 isoforms [38,39,40].

No matter whether being AMPK-dependent or independent, metabolic effects of AICAr may be of relevance for the potential treatment of type 2 diabetes [41]. AICAr induces hypoglycemia in vivo [42,43] and the effect is abolished in mice lacking AMPK [32,33,35], suggesting that the effect can be more ascribed to AMPK-dependent entry of glucose than to AMPK-independent effects of AICAr on the inhibition of gluconeogenesis. In addition, AICAr may help to reduce peripheral resistance to insulin action because AICAr acts to reduce the storage of fatty acids in adipose tissue [37]. In various animal models of insulin resistance, AICAr administration has been shown to improve metabolic disturbances and to enhance insulin sensitivity in peripheral tissues [44,45,46,47]. Systemic AICAr administration in humans exerted beneficial effects by reducing hepatic glucose output and increasing glucose uptake in skeletal muscle [43,48]. However, as shown in Table 2, AICAr has shown a very poor oral bioavailability in clinical trials [49]. The preferred route of administration is through continuous intravenous injection and that renders it quite unsuitable for chronic treatment of metabolic disorders like diabetes.

**Table 1 cells-10-01095-t001:** AMPK-dependent and AMPK-independent effects of AICAr in transgenic mice models.

Tissue	AICAr Effect	AMPK-Dependency	Model
**Skeletal muscle**	Increased glucose uptake	AMPK-dependent	Skeletal and cardiac muscle-specific expression of ampkα2-KD [32]
			Total ampkα2-KO [33]
			Total ampky3-KO [34]
			Muscle-specific expression of ampkα2-DN [35]
	Increased FAO	AMPK-independent	Muscle-specific ampkα2-KD [50]
	Increased mrna HKII and (PGC)-1α	AMPK-dependent	Total ampkα2-KO [51]
**Liver**	Decreased glucose phosphorylation	AMPK-independent	Liver-specific ampkα1α2-KO [39]
	Decreased gluconeogenesis	AMPK-independent	Liver-specific ampkα1α2-KO [38]
	Inhibited OXPHOS	AMPK-independent	Liver-specific ampkα1α2-KO [40]
	Decreased lipogenesis	AMPK-dependent	Liver-specific ampkα1α2-KO [36]
	Increased FAO	AMPK-dependent	Liver-specific ampkα1α2-KO [36]
**Adipose tissue**	Decreased FA synthesis	AMPK-dependent	Adipose tissue-specific ampkα1/α2-KO [37]
	Decreased lipolysis	AMPK-dependent	AMPK α1-KO [52]
**Whole body**	Acute hypoglycemia	AMPK-dependent	Skeletal and cardiac muscle-specific ampkα2-KD [32]
			Total ampkα2-KO [33]
			Muscle-specific ampkα2-DN [35]

DN, dominant negative; FA, fatty acid; FAO, fatty acid oxidation; HKII, hexokinase II; KD, kinase dead; KO, knockout; OXPHOS, oxidative phosphorylation; (PGC)-1α, peroxisome-proliferator-activated receptor γ coactivator.

## 3. AICAr, Adenosine, and Ischemic Heart

The first study of the safety and tolerance of AICAr was done in 1991, much before the recognition of AICAr as an AMPK agonist to establish pharmacokinetics of a drug that raised interest as a novel adenosine-regulating agent [49]. Adenosine is a potent vasodilator that plays a key role in reducing ischemia/reperfusion injury, but the applications for systemic adenosine are limited owing to peripheral hemodynamic actions [13]. As shown in Figure 1, AICAr shares structural similarities with adenosine, and therefore, can increase the extracellular concentrations of adenosine by competing for the nucleoside transporter [20]. In addition, AICAR increases intracellular concentrations by inhibiting adenosine deaminase and increasing the production of adenosine rather than inosine from ATP catabolism. Several animal studies performed in the 1980s demonstrated that AICAr or acadesine infusion improved postischemic recovery in the heart [53,54], and prompted the first international randomized studies in human participants undergoing coronary artery bypass graft surgery (CAGS). As shown in Table 2, several clinical studies testing the effects of AICAr were performed [16,18,19], and outcomes of these studies were examined in a 1997 meta-analysis that revealed that AICAr can reduce early cardiac death, myocardial infarction, and combined adverse cardiovascular outcomes [14]. However, these promising meta-analysis results were not confirmed by later clinical trials. In 2012, the RED-CABG trial was stopped early after interim data failed to indicate a reduction in morbidity or mortality among intermediate- to high-risk patients receiving AICAr versus placebo [15]. 

Before the mode of action via AMPK was appreciated, AICAr-mediated protection of myocardium was ascribed only to the effects of adenosine on vasodilation and inhibition of platelet aggregation and neutrophil activation [13,54]. Later studies provided the link between the activation of AMPK and AICAr-mediated effects on glucose and glycogen metabolism in heart muscle [30,55]. Although several issues have complicated the interpretation of pharmacological manipulation of AMPK by AICAR in cardiomyocytes [56], AICAr has been described to exert several beneficial effects, including improved cardiac function in a model of endotoxin-induced myocardial inflammation [57], prevention of cardiac fibrosis [58], and metabolic changes in cardiomyocytes exposed to free fatty acids that could protect the hearts of patients with type 2 diabetes against ischemia-reperfusion injury [59]. 

In human aortic endothelial cells, AICAr stimulated AMPK activity and nitric oxide (NO) production, and the effects were proved to be AMPK-dependent since the effects were inhibited by the expression of a dominant-negative (DN) AMPK mutant [60]. Similar AMPK-dependent effects on NO production were observed in response to hypoxia [61], and studies performed in the knockout of the upstream kinase LKB1 confirmed the important role of AMPK in angiogenesis [62]. 

**Table 2 cells-10-01095-t002:** AICAr in clinical trials.

Year	Condition	Trial type	Doses	Toxicity	Outcome
1991	Healthy volunteers	Phase I	PO and IV: 10, 25, 50, and 100 mg/kg	Well tolerated, only mild and transient side effects	Poor oral bioavailability, the rapid decline of post-infusion plasma concentrations [49]
1994	Lesch-Nyhan Syndrome	Case Report	PO: 30 mg/kg/day for 4 days followed by 100 mg/kg/day for 4 days	No adverse events	No changes in plasma levels of AICAr, confirm the estimate of <5% oral bioavailability of AICAr in humans [63]
1994	Coronary artery bypass grafting	Multicenter RCT, Phase II	Continuous IV for 7 h: 0.19 and 0.38 mg/kg/min	Well tolerated, mild hyperuricemia	Limits the severity of post-bypass myocardial ischemia at higher dose [16]
1994	Exercise-induced myocardial ischemia in patients with chronic stable angina pectoris	Single center RCT, Phase II	IV: 6–48 mg/kg	Well tolerated, mild asymptomatic hyperuricemia, mild asymptomatic hypoglycemia	No significant difference in comparison to placebo [17]
1995	Coronary artery bypass grafting	Multicenter RCT, Phase II	Continuous IV for 7h: 0.1 mg/kg/min; in cardioplegic solution 5 mg/mL	No adverse events	No significant difference in MI in the overall study group; significantly reduced the incidence of Q-wave MI in high-risk patients [18]
1995	Coronary artery bypass grafting	Multicenter RCT, Phase II	Continuous IV for 7 h: 0.05 and 0.1 mg/kg/min	Well tolerated, mild hyperuricemia	No significant difference in comparison to the placebo may reduce the incidence of larger Q-wave MI [19]
1997	Coronary artery disease	Single center RCT, Phase II	Continuous IV: 5, 10, 20, 50 mg/kg	Well tolerated, hyperlactacidemia	At higher doses, minor beneficial effects on ejection fraction and myocardial lactate metabolism were observed [8]
2006	Coronary artery bypass grafting	Multicenter RCT, Phase III	Continuous IV for 7 h: 0.1 mg/kg/min	No adverse events	Reduces the severity of acute post-reperfusion MI, substantially reducing the risk of dying over the 2 years after infarction [9]
2007	Healthy volunteers	Phase I	Continuous IV for 3 h: 10 mg/kg/h	No adverse events	Acutely stimulates muscle 2-DG uptake with a minor effect on whole-body glucose disposal [43]
2009	Healthy volunteers	Phase I	Continuous intra-arterial infusion for 110 min: 1, 2, 4, or 8 mg/min/dL forearm tissue	No adverse events	Potent vasodilation in the skeletal muscle vascular bed; does not increase skeletal muscle glucose uptake [64]
2012	Coronary artery bypass grafting	Multicenter RCT, Phase III	Continuous IV for 7 h: 0.1 mg/kg/min	No adverse events	No significant reduction in the composite of all-cause mortality, nonfatal stroke, or severe left ventricular dysfunction through 28 days [15]
**Hematologic malignancies**					
2013	Relapsed/refractory chronic lymphocytic leukemia (CLL)	Multicenter open-label clinical study, Phase I/II	Continuous IV: single doses of 50–315 mg/kg; two doses at 210 mg/kg; five doses at 210 mg/kg	Grade ≥2 hyperuricemia (not clinically significant), transient anemia and/or thrombocytopenia (not clinically significant), renal impairment, and transient infusion-related hypotension (clinically significant)	210 mg/kg was the MTD and OBD. Multiple-dose administrations at the OBD have an acceptable safety profile [6]
2019	Azacytidine refractory MDS/AML patients	Phase I/II	Continuous IV: 140 mg/kg or 210 mg/kg	Trial stopped after 2 to 3 cycles due to serious renal toxicities	Side effects of high doses preclude its use in patients with strong comorbidities; one patient exhibited a very strong reduction (50%) of his blast count after only 2 cycles of AICAr and more than 70% after 6 cycles [10]

2-DG, 2-Deoxyglucose; IV, intravenous; MI, myocardial infarction; MTD, maximum tolerated dose; OBD, optimal biological dose; PO, *per os*.

## 4. AICAr as an “Exercise in a Pill”

In 2008, Narkar et al. reported that, even in sedentary mice, 4 weeks of AICAr treatment alone enhanced running endurance by 44% and induced genes linked to oxidative metabolism in muscle cells. AICAr induced fatigue-resistant type I (slow-twitch) fiber specification, and AMPK activation by AICAr was sufficient to increase running endurance without additional exercise signals [65]. As would be expected, AICAr was immediately set into the limelight of not only the scientific but sports and wider community as a new exercise mimetic or an “exercise in a pill.” In less than a year, the French anti-doping agency raised concerns about a new metabolic substance being used on Tour de France, and AICAr soon appeared on the list of substances banned by the World Anti-Doping Agency [66]. In 2012, a sports doctor and nine others of the Spanish cycling team were arrested in connection with an international network supplying the synthetic AMPK activator AICAR as a “next generation superdrug” performance-enhancing drug [67].

As shown in Table 1, the majority of the effects of AICAr on skeletal muscles are AMPK-dependent. AICAr-induced glucose uptake in skeletal muscle was abolished in the knockout of the α 2 [32,33,35] and α 3 isoforms of AMPK [34]. Both AICAr and treadmill exercise increased insulin sensitivity to stimulate glucose uptake, and these effects were not observed in mice with reduced or ablated AMPK activity in skeletal muscle [68,69]. However, the mechanisms of exercise- and AICAr-mediated glucose transport diverge at some point downstream of AMPK since AICAr-induced effects were absent in muscle-specific knockout of atypical PKC, and atypical PKC was not required in treadmill exercise [70]. Both AICAr and exercise induce AMPK activation and metabolic stress, but the mechanical stress is only caused by exercise, so that the combination of two may be useful in some conditions. In chronic inflammatory myopathy model mice, the combination of AICAr and exercise reverse apoptosis of fibro-adipogenic progenitors and improves muscle function and regeneration [70]. To add another layer of intersection between the exercise and AICAr, a recent study of daytime variance in exercise capacity revealed that exercise itself may induce an increase in the level of endogenous ZMP (AICA ribotide or AICAR). Moreover, endogenous ZMP was induced by exercise in a time-dependent manner and had the same effects as exogenous AICAr on AMPK activation, glycolysis, and fatty acid oxidation [71]. 

There is an increasing number of papers focusing on the effect of aging on mitochondrial biogenesis and the beneficial effects of exercise and exercise mimetic in delaying aging-mediated decline in AMPK activity. Reznick et al. examined AMPK activity in young and old rats and found that acute stimulation of AMPK α 2 activity by AICAr and exercise was blunted in skeletal muscle of old rats, suggesting that reductions in AMPK activity may be an important contributing factor in aging-associated reductions in mitochondrial biogenesis and dysregulated lipid metabolism [72]. However, 500 mg/kg of AICAr has beneficial effects on cognition and motor coordination in both young and old mice, and these effects were absent in the muscle-specific mutated AMPK α 2-subunit, providing support for a muscle-mediated mechanism [73]. In another study, the same AICAr concentration had no beneficial impact on skeletal muscle of wild-type mice and could not be considered as a mimetic of exercise in aged mice, but improved aerobic running capacity and mitochondrial function in aged myostatin KO mice [74]. In a mouse model for Duchene muscular dystrophy, AICAr treatment enhanced the effects of exercise [75] and improved muscle function, probably by stimulating autophagy [76]. 

## 5. AICAr, AMPK, Proliferation and Cell Cycle

As a central metabolic regulator that reacts to an increase in AMP/ATP ratio, AMPK restricts growth and proliferation in response to energetic or nutritional stress. AICAr was first reported to suppress protein synthesis in rat skeletal muscle through down-regulation of the mechanistic target of rapamycin (mTOR) signaling, including p70 S6 Kinase 1 (S6K1) and eukaryotic translation initiation factor 4E binding protein 1 (4E-BP1), which are involved in the regulation of protein translation [29]. As shown in Figure 2, mTOR is a catalytic subunit of two functionally distinct protein complexes, mTORC1 (mTOR complex 1) and mTORC2 (mTOR complex 2), and both S6K1 and 4E-BP1 lie downstream of mTORC1. AICAr inhibits mTORC1 but activates mTORC2, and both effects are AMPK-dependent. AICAr-mediated inhibition of mTORC1 requires the phosphorylation of protein raptor [77]. AICAr-mediated activation of mTORC2 did not result from AMPK-mediated suppression of mTORC1, and thus, reduced negative feedback on phosphatidylinositol 3-kinase (PI3K) flux, but rather on direct phosphorylation of mTOR in complex with rictor and phosphorylated Akt as a downstream target [78]. 

While mTORC1 inhibition ensures restriction of cell growth, stabilization of p53, another major target for activated AMPK, causes cell cycle arrest [79]. In many cell types, AICAr-mediated inhibition of proliferation occurs concomitantly with AMPK phosphorylation and is associated with inhibition of mTOR activity and/or p53 accumulation [80,81,82,83,84,85,86]. However, to prove that AICAr effects are dependent on AMPK, small interference RNAs targeted at AMPK or the expression of constitutively active or DN AMPK mutants should be used to test for the role of AMPK in the effects of AICAr [87]. In addition, if AICAr effects are truly AMPK-dependent, they should be mimicked by other AMPK agonists, and particularly with AMPK agonists which are more specific AMPK activators, like A-769662 [86], MK-8722 [88], and PF-739 [89]. Another more selective agonist is C13, a prodrug that is converted by cellular metabolism into C2, an AMP analog whose binding site overlaps with that of AMP and AICAR, but is orders of magnitude more potent in AMPK activation [90]. 

Using these approaches, AICAr-induced proliferative arrest of primary mouse embryonic fibroblasts (MEF) [81], various cancer cell lines [82], and aneuploid MEF cells [84] were shown to be AMPK-dependent as the effects were more or less inhibited in cells infected with a retroviral vector expressing DN AMPK. However, in MEF cells lacking both catalytic isoforms of AMPK, AICAr effects on proliferation and cell cycle arrest were AMPK-independent, and the effects on apoptosis were even more pronounced, while A-769662 exhibited specific AMPK-dependent effects on cell growth and metabolism [86]. In glioma cells, AICAr inhibited proliferation in vitro and in vivo independent of AMPK, while A-769662 had no effects on proliferation [91], and similar AMPK-independent effects of AICAr on proliferation have been observed in various leukemia cells [92,93].

The cell cycle analyses of AICAr-arrested cells in some studies revealed an increase in the proportion of cells in the G_0_/G_1_ phase, as would be expected from the mechanism of cell cycle arrest in response to AMPK activation and mTORC1 inhibition [23]. However, in embryonic stem cells, AICAr increased the cell population at both G_1_ and non-cycling S phases [85]. Furthermore, an arrest in the S phase has been observed in MEFs [86], cancer cell lines [94], and leukemia cells [95]. In glioma cell lines, AICAr-induced an AMPK-independent arrest in the G_2_/M phase [91]. To understand the mechanism responsible for proliferation arrest in response to AICAr, we have to go back to the role of endogenous AICAR in de novo purine synthesis that has been well-known to affect cell growth much before the discovery of the role of AICAr in AMPK activation. 

## 6. AICAr, ZMP, and Purine Synthesis

As shown in Figure 3, AICA ribotide (AICAR) or ZMP is a normal cellular intermediate in de novo purine synthesis. AICAR or ZMP is increased in Lesch-Nyhan syndrome, one of the most common disorders of purine and pyrimidine metabolism. The Lesch-Nyhan results from a deficiency of hypoxanthine-guanine phosphoribosyltransferase (HGPRT), so that the activity of the salvage pathway is diminished and the de novo pathway of purine nucleotide synthesis accelerated, leading to an accumulation of ZMP or AICAR [96]. Recent studies reported an increase in the levels of AICAr, which normally arises from dephosphorylation of ZMP, in urine and cerebrospinal fluid of patients with deficiency of HGPRT [97], suggesting that the accumulation of ZMP may be the cause of neurological abnormalities, similar to the toxicity of ZMP accumulation in yeast [98]. Yeast is a good experimental system to study the effects of AICAr that are AMPK-independent as the yeast AMPK orthologue SNF1 is activated by ADP rather than AMP, and genes strongly regulated by Snf1p are not identical to AICAr-regulated transcription. In the yeast model, disruption of nucleotide homeostasis was identified as a crucial feature of AICAr toxicity [99], suggesting the similar role of nucleotide metabolism in AMPK-independent growth arrest induced by an exogenous AICAr in human cell lines. 

When two acute lymphoblastic leukemia (ALL) cell lines were exposed to AICAr and A-769662, metabolomics analysis revealed that AICAr-treated cells excreted significant amounts of metabolites involved in purine and pyrimidine metabolism, and these metabolites were not detected in the samples treated with A-769662. AICAr-mediated increase in the level of hypoxanthine, adenine, guanine, and inosine can be ascribed to an increase of the purine metabolism generated by AICAr entering these pathways. Moreover, pyrimidine metabolism was also significantly affected by AICAr, and an increase in the levels of orotate, dihydroorotate, and carbamoylaspartate suggested a possible inhibition at the level of UMP synthase [100]. A possible role for inhibition of UMP synthase and pyrimidine starvation in AICAr-mediated apoptosis was further corroborated with metabolomics screen in multiple myeloma cells. In these cells, AICAr induced an accumulation in the S phase, an increase in purine, a decrease in pyrimidine metabolites, and the most striking increase in the level of orotate [101]. The same effects on the levels of purine and pyrimidine metabolites were observed in AICAr-treated ALL cell lines, in which AICAr-mediated arrest in the S phase was AMPK-independent, but attenuated with the addition of uridine [102]. The inhibition of pyrimidine synthesis at the level of UMP synthase was ascribed to the lack of phosphoribosyl pyrophosphate (PRPP), and a decrease in PRPP in response to AICAr has been described in other model systems [103,104]. 

However, most of the drugs that increase the level of endogenous ZMP act to activate AMPK so that it is difficult to completely rule out the possible involvement of AMPK in antiproliferative effects. An inhibitor of AICAR transformylase (AICART), an enzyme that catalyzes the last two steps of purine de novo synthesis and metabolizes AICAR, induces an increase in the level of AICAR or ZMP, and endogenous ZMP was capable of activating AMPK and its downstream signaling pathways [105]. The antifolate pemetrexed inhibits the folate-dependent enzyme in de novo purine biosynthesis, increases ZMP, and activates AMPK [106]. Methotrexate, a well-known cytostatic drug, inhibits purine de novo synthesis and potentiates the ability of exogenous AICAr to increase the level of ZMP by inhibiting AICART (Figure 3). Consequently, methotrexate enhances the ability of AICAr to activate AMPK and to inhibit the growth of human cancer cell lines [107], and promote glucose uptake and lipid oxidation in skeletal muscle [108]. 

## 7. AICAr, AMPK, Cancer, and Leukemia

In 2003, Campas et al. reported that AICAr activates AMPK and induces apoptosis in primary samples of B-cell chronic lymphocytic leukemia (CLL) in vitro [11]. Two years later, an epidemiological study revealed that metformin, another AMPK activator had a protective role in the development of cancer, and thus, invigorated interest in the possible use of AMPK agonists in the treatment of cancer [109]. However, although it would be expected that AMPK, acting as a downstream target of LKB1, has tumor suppressor activity, the results from many studies point that the role of AMPK in cancer is much more complex; AMPK may suppress tumor growth before tumorigenesis, but once cancer has arisen, AMPK may instead support the survival of cancer cells [23]. In the meantime, many other studies described the beneficial effects of AICAr, especially in hematological malignancies, and most of these effects turned out to be AMPK-independent.

AICAr-induced apoptosis in CLL cells was independent of AMPK since AMPK activation with phenformin or A-769662 failed to induce apoptosis, and AICAr had effects in B lymphocytes from AMPKα1(−/−) mice [92]. The effects of AICAr on CLL cells in vitro were demonstrated at doses that are tolerated well when achieved in plasma after intravenous injection [11] so that the multicenter study testing the effects of AICAr for patients with relapsed/refractory CLL had been initiated. The results of phase I/II revealed that AICAr had an acceptable safety profile and antileukemic activity in patients with poor prognosis [6]. 

Cytotoxic effects have been reported in other hematological malignancies. AICAr-induced apoptosis and concurrent activation of AMPK were described in childhood acute lymphoblastic leukemia (ALL) cell lines [110], as well as in B cells isolated from patients with mantle cell lymphoma and splenic marginal zone lymphoma [7]. However, the role of AMPK in cytotoxic effects of AICAr was not investigated. In chronic myelogenous leukemia (CML) cell lines [12] and primary samples [111], AICAr had antiproliferative effects, but AMPK knock-down by shRNA failed to prevent the effect of AICAr, indicating an AMPK-independent mechanism [12]. In azacytidine (Aza)-resistant myelodysplastic syndrome and acute myeloid leukemia (MS/AML) cell lines and primary samples, 2 mM AICAr blocked proliferation, and these initial findings led to a phase I/II clinical trial using AICAr in 12 patients with Aza-refractory MDS/AML patients. Despite a very good response in one out of four patients, the trial was stopped because the highest dose of AICAr caused serious renal side effects in patients with severe comorbidities [10].

Our recent studies demonstrated that lower concentrations of AICAr alone are capable of inducing differentiation of various monocytic AML cell lines, as well as a subset of primary blasts isolated from the bone marrow of AML patients [93,95,112]. These effects were not mimicked by metformin [93] or A-769662 [113]. Differentiation therapy of acute promyelocytic leukemia, a particular subtype of AML, with all-trans retinoic acid provides the most successful pharmacological therapy of AML so that the approach of treating cancer cells by differentiation instead of killing is well appreciated, especially after the recent success of AML differentiation with inhibitors of mutated isocitrate dehydrogenase [114]. Of note, differentiation in response to AICAr has been described in various other cell systems, including erythroid differentiation of embryonic stem cells [85], mineralization of osteoblastic MC3T3-E1 cells [115], and astroglial differentiation of neural stem cells [116], but the mechanisms have not been completely elucidated. In our model of monocytic leukemia cells, the effects of AICAr were not AMPK-dependent, since both growth arrest and differentiation were preserved in AML cells with siRNA-downregulated AMPK [93]. In CML cell lines, an AMPK-independent cell death involved AICAr-induced autophagy [12], but our study showed that, although AICAr induced autophagy in parallel with differentiation, AICAr-mediated differentiation did not depend on the presence of key proteins of canonical autophagy pathway [113]. However, AICAr-induced differentiation was inhibited by downregulation of checkpoint kinase 1 (Chk1), and activation of Chk1 was induced by defects in pyrimidine synthesis [95]. As shown in Figure 3, AICAr inhibits pyrimidine synthesis at the level of UMP-synthase, and thus, shares the same mechanism of leukemia differentiation as the one that has been recently described in response to brequinar, a well-known inhibitor of dihydroorotate dehydrogenase [117]. These results further corroborate the hypothesis that antiproliferative effects of AICAr are more related to the effects of AICAr on nucleotide synthesis and cell cycle arrest than on the activation of AMPK that is usually measured to be increased concomitantly.

## 8. Conclusions

Over the last 25 years, AICAr has been used in hundreds of studies as an activator of AMPK. The results of these initial studies pointed to the important roles of AMPK, and many of them have been later confirmed by studies in transgenic mice or by using models of cells with overexpression or down-regulation of AMPK. However, AICAr accumulates in cells in millimolar concentrations and exerts many AMPK-independent or “off-target“ effects so that allowances must be made for the possible use of AICAr. Although AICAr is no longer recommended as a specific AMPK agonist [118], mostly because there are many more specific activators of AMPK available nowadays, AICAr can be still useful in an initial screen to test for AMPK activation, especially when combined with other AMPK agonists and proper methods for AMPK downregulation. In addition, AICAr is still a highly promising pharmacological agent having many beneficial effects in metabolism, hypoxia, exercise, and cancer.

## Figures and Tables

**Figure 1 cells-10-01095-f001:**
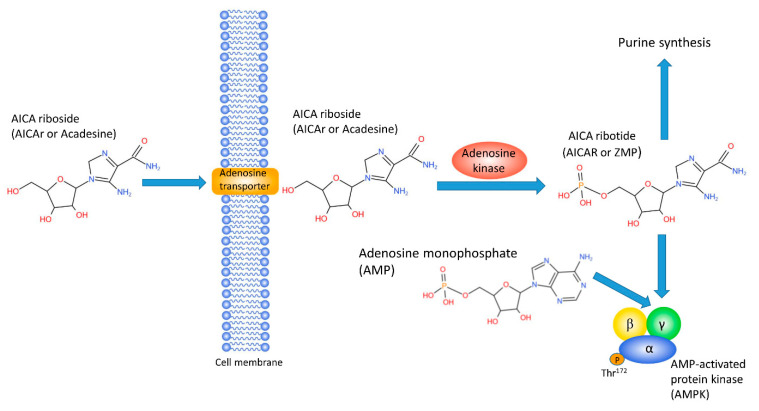
AICAr structure and mechanism.

**Figure 2 cells-10-01095-f002:**
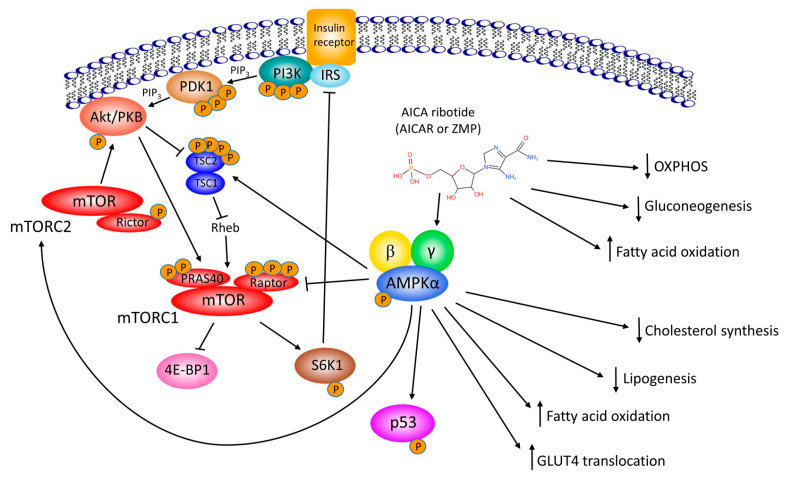
A schematic presentation of molecular mechanisms of AICAr Akt/PKB, Akt/protein kinase B; PDK1, 3-phosphoinositide dependent protein kinase-1; PI3K, phosphatidylinositol 3-kinase; PIP3, phosphatidylinositol (3,4,5)-trisphosphate; IRS, insulin receptor substrate; TSC1/2, tuberous sclerosis complex 1 protein / tuberous sclerosis complex 2 protein; Rheb, Ras homolog enriched in the brain; mTOR, mechanistic target of rapamycin; mTORC1, mTOR complex 1; mTORC2, mTOR complex 2; Rictor, rapamycin-insensitive companion of mTOR; PRAS40, proline-rich Akt substrate 40; Raptor, regulatory-associated protein of mTOR; AMPK, adenosine monophosphate (AMP)-activated protein kinase; p53, tumor suppressor protein p53; S6K1, p70 S6 Kinase 1; OXPHOS, oxidative phosphorylation; 4E-BP1, eukaryotic translation initiation factor 4E binding protein 1; GLUT4, glucose transporter 4.

**Figure 3 cells-10-01095-f003:**
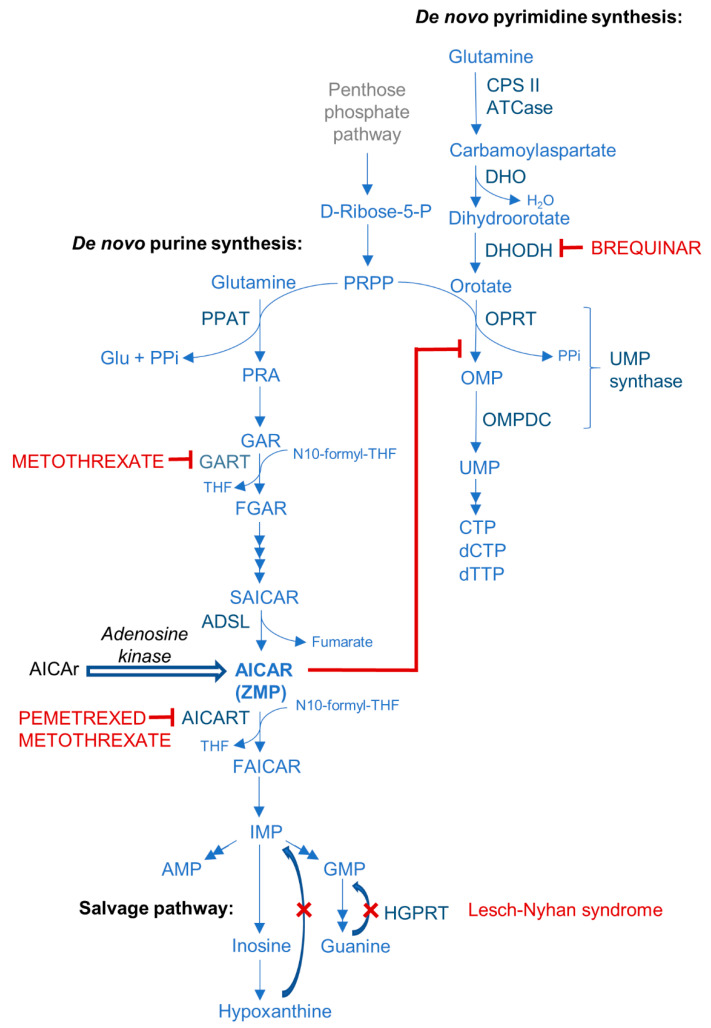
Schematic representation of AICAR involvement in purine and pyrimidine synthesis. D-Ribose-5-P, ribose 5-phosphate; PRPP, 5-phosphoribosyl 1-pyrophosphate; PPAT, phosphoribosyl pyrophosphate amidotransferase; Glu, glutamate; PPi, pyrophosphate; PRA, 5-phosphoribosylamine; GAR, 5'-phosphoribosylglycinamide; N10-formyl-THF, 10-formyl tetrahydrofolate; GART, glycinamide ribonucleotide transformylase; THF, tetrahydrofolate; FGAR, 5'-phosphoribosyl-N-formylglycinamide; SAICAR, 5'-phosphoribosyl-5-aminoimidazole-4-N-succinocarboxamide; ADSL, adenylosuccinate lyase; AICAr, 5-amino-4-imidazolecarboxamide (AICA) ribonucleoside; AICAR, 5-amino-4-imidazolecarboxamide (AICA) ribonucleotide; AICART, AICAR transformylase; FAICAR, 5-formamidoimidazole-4-carboxamide ribotide; IMP, inosine monophosphate; AMP, adenosine monophosphate; GMP, guanosine monophosphate; HGPRT, hypoxanthine-guanine phosphoribosyltransferase; CPS II, carbamoyl phosphate synthetase II; ATCase, aspartate transcarbamoylase; DHO, dihydroorotase; DHODH, dihydroorotate dehydrogenase; OPRT, orotate phosphoribosyltransferase; OMP, orotidine-5'-monophosphate; OMPDC, orotidine 5'-phosphate decarboxylase (OMP decarboxylase); UMP, uridine monophosphate; CTP, cytidine triphosphate; dCTP, deoxycytidine triphosphate; dTTP, deoxythymidine triphosphate.

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
