# Peer review of "AICAr, a Widely Used AMPK Activator with Important AMPK-Independent Effects: A Systematic Review"

_cells, 2021, doi:10.3390/cells10051095_

Round 1
Reviewer 1 Report
This is a useful, thorough, and generally well-written, review on the AMPK-dependent and AMPK-independent effects of AICAr, by authors who have made contributions to the field. I found Tables 1 and 2, and the later section on leukemia and cancer, to be particularly useful. I have mostly relatively minor comments only:
MAJOR POINTS:
- Line 186: the authors do a good job of pointing out that AICAr is a rather non-specific activator of AMPK, and that more selective agonists are now available. On line 186 they mention A-769662, which is also referred to elsewhere in the review. However, A-769662 also has AMPK-independent effects, and much more potent agonists are now available, incliuding MK-8722 and PF-739 (which bind to the same site as A-769662, and unlike the latter also activate AMPK complexes containing either beta-1 or beta-2). Another is C13, a prodrug that is converted by cellular metabolism into C2. Although C2 is another AMP analogue whose binding site overlaps with that of AMP and AICAR/ZMP [Langendorf et al (2016) Nat. Comm. 7:10912] and it is also highly selective for AMPK complexes containing the alpha-1 isoform, it is orders of magnitude more potent than either AICAR or AMP. I think that these other agents should be mentioned as better alternatives to AICAr if the aim of the study is to activate AMPK.
- Line 64: an important omission from this review would seem to be any mention of the work from Sakamoto's lab showing that AICAR mimics the effect of AMP on fructose-1,6-bisphosphatase [e.g. Hunter et al (2018) Nat. Med. 24:1395]. This is likely to largely explain the effects of AICAr on gluconeogenesis in the liver, and must be included.
MINOR POINTS:
- Lines 42-43: although they are given in ref. 21, I think that citations to studies showing that "AICAr enters the cells through adenosine transporters and becomes phosphorylated by adenosine kinase into AICAR" should be added.
- Line 43: yes, AICAR/ZMP is 40- to 50-fold less potent than AMP, as reported in ref. 1. It is worth stressing the low potency of AICAR on AMPK, and the fact that it can accumulate to mM concentrations in cells (also in ref. 1), to emphasize the point that it was always likely to have AMPK-independent effects.
- Line 89: I would not describe AICAr as an "AMPK mimetic", although "AMPK agonist" is probably OK.
- Lines 93-95: a citation is needed here [e.g. Gadalla et al (2004) J. Neurochem. 88:1272].
- Line 272: ref. 100 was a study of all cancers, not just colon cancer.
Author Response
Thank you very much for your comments on our manuscript. We have addressed all comments and made every attempt to incorporate your suggestions as thorough as possible. All revisions are clearly highlighted using the "Track Changes" function.
The answers to the comments are as follows:
Reviewer 1: This is a useful, thorough, and generally well-written, review on the AMPK-dependent and AMPK-independent effects of AICAr, by authors who have made contributions to the field. I found Tables 1 and 2, and the later section on leukemia and cancer, to be particularly useful. I have mostly relatively minor comments only:
MAJOR POINTS:
Line 186: the authors do a good job of pointing out that AICAr is a rather non-specific activator of AMPK, and that more selective agonists are now available. On line 186 they mention A-769662, which is also referred to elsewhere in the review. However, A-769662 also has AMPK-independent effects, and much more potent agonists are now available, including MK-8722 and PF-739 (which bind to the same site as A-769662, and unlike the latter also activate AMPK complexes containing either beta-1 or beta-2). Another is C13, a prodrug that is converted by cellular metabolism into C2. Although C2 is another AMP analogue whose binding site overlaps with that of AMP and AICAR/ZMP [Langendorf et al (2016) Nat. Comm. 7:10912] and it is also highly selective for AMPK complexes containing the alpha-1 isoform, it is orders of magnitude more potent than either AICAR or AMP. I think that these other agents should be mentioned as better alternatives to AICAr if the aim of the study is to activate AMPK.
We highly appreciate the comment. As suggested, more selective agonists and appropriate references are added, like: ”…MK-8722 [88], and PF-739 [89]. Another more selective agonist is C13, a prodrug that is converted by cellular metabolism into C2, an AMP analogue whose binding site overlaps with that of AMP and AICAR, but is orders of magnitude more potent in AMPK activation [90]…”.
Line 64: an important omission from this review would seem to be any mention of the work from Sakamoto's lab showing that AICAR mimics the effect of AMP on fructose-1,6-bisphosphatase [e.g. Hunter et al (2018) Nat. Med. 24:1395]. This is likely to largely explain the effects of AICAr on gluconeogenesis in the liver, and must be included.
As suggested, the work from Sakamoto's lab showing that AICAR mimics the effect of AMP on fructose-1,6-bisphosphatase is included: “…and fructose-1-6-biphosphatase, a rate controlling enzyme in gluconeogenesis, which is likely to explain why the effects of AICAr on gluconeogenesis in the liver are AMPK-independent [31].“.
MINOR POINTS:
Lines 42-43: although they are given in ref. 21, I think that citations to studies showing that "AICAr enters the cells through adenosine transporters and becomes phosphorylated by adenosine kinase into AICAR" should be added.
As suggested, appropriate references are added: ref. 20 (Gadalla et al., J Neurochem 2004) and 21 (Hawley et al., Cell Chem Biol 2020).
Line 43: yes, AICAR/ZMP is 40- to 50-fold less potent than AMP, as reported in ref. 1. It is worth stressing the low potency of AICAR on AMPK, and the fact that it can accumulate to mM concentrations in cells (also in ref. 1), to emphasize the point that it was always likely to have AMPK-independent effects.
As suggested, this is now included in lines: “…AICAR or ZMP activates AMPK but it is 40- to 50- fold less potent than AMP in AMPK activation and accumulates in high concentrations in cytoplasm [1], so that it was always likely that AICAr may have several AMPK-independent effects…”.
Line 89: I would not describe AICAr as an "AMPK mimetic", although "AMPK agonist" is probably OK.
We apologize for the mistake; “AMPK mimetic“ is replaced with “AMPK agonist“.
Lines 93-95: a citation is needed here [e.g. Gadalla et al (2004) J. Neurochem. 88:1272].
As suggested, a citation is included in lines: “…by competing for the nucleoside transporter [20].”.
Line 272: ref. 100 was a study of all cancers, not just colon cancer.
We greatly appreciate this comment and apologize for the mistake; “colon“ is deleted.
We highly appreciate your valuable comments and careful evaluation of our work and hope that this revision meets with your approval.
Sincerely yours,
Dora Visnjic
Reviewer 2 Report
In this review, authors intend to summarize the current understanding on AMPK-dependent and independent effects of AICAr during various vital processes. The manuscript has been well presented that might be of interest to particular section of research community, however, addition of few more information as outlined below will provide deeper mechanistic insights on AICAr and considerations for its uses in future studies.
Comments:
1) A schematic presentation of molecular mechanisms of AICAr discussed in this review should be provided.
2) The theme of this review focusses on AMPK independent effects of AICAr. Therefore, more molecular mechanistic details on AMPK independent effects of AICAr such as decreased gluconeogenesis or OXPHOS should be elaborated. The authors should provide more mechanistic details on effects of AICAr on mTOR, including direct or indirect effects. Whether AICAr modulated both mTORc1 or mTORc2 to the same extent through AMPK dependent or independent mechanisms.
3) The authors should discuss limitations for using AICAr, in both in vitro or in vivo models, and recommendations or caution for future studies when using AICAr should be provided.
Author Response
Thank you very much for your comments on our manuscript. We have addressed all comments and made every attempt to incorporate your suggestions as thorough as possible. All revisions are clearly highlighted using the "Track Changes" function.
The answers to the comments are as follows:
Reviewer 2: In this review, authors intend to summarize the current understanding on AMPK-dependent and independent effects of AICAr during various vital processes. The manuscript has been well presented that might be of interest to particular section of research community, however, addition of few more information as outlined below will provide deeper mechanistic insights on AICAr and considerations for its uses in future studies.
Comments:
1) A schematic presentation of molecular mechanisms of AICAr discussed in this review should be provided.
As suggested, a schematic presentation of molecular mechanisms of AICAr is included now as a new Figure 2 (R).
2) The theme of this review focusses on AMPK independent effects of AICAr. Therefore, more molecular mechanistic details on AMPK independent effects of AICAr such as decreased gluconeogenesis or OXPHOS should be elaborated. The authors should provide more mechanistic details on effects of AICAr on mTOR, including direct or indirect effects. Whether AICAr modulated both mTORc1 or mTORc2 to the same extent through AMPK dependent or independent mechanisms.
As suggested, we have introduced a new reference regarding the mechanism of AICAr-mediated decrease in gluconeogenesis (Hunter et. al., Nat. Med. 2018) described in lines: “…and fructose-1-6-biphosphatase [31], a rate controlling enzyme in gluconeogenesis, which is likely to explain why the effects of AICAr on gluconeogenesis in the liver are AMPK-independent…“, and introduced a ref. 40 (Guigas et al., Biochem J 2007) in Table 1 (R) and text regarding the effects of AICAr on OXPHOS. Both effects on gluconeogenesis and OXPHOS are included in new Figure 2 as AMPK-independent effects of AICAr. The effects of AICAr on mTORC1 and mTORC2 are described in new section of paragraph 1, subheading “AICAr, AMPK, proliferation and cell cycle“ and shown in a new Figure 2 (R).
3) The authors should discuss limitations for using AICAr, in both in vitro or in vivo models, and recommendations or caution for future studies when using AICAr should be provided.
As suggested, we recommended caution for future studies in Conclusion section: “…However, AICAr accumulates in cells in millimolar concentrations and exerts many AMPK-independent or „off-target“ effects so that allowances must be made for the possible use of AICAr…”.
We appreciate your careful evaluation of our work and hope that this revision meets with your approval.
Sincerely yours,
Dora Visnjic
Reviewer 3 Report
The manuscript (Manuscript ID: cells-1164864) entitled “AICAr, a widely used AMPK activator with important AMPK-independent” was by Dora Višnjić, et al. The authors reviewed the present knowledge on AMPK-dependent and AMPK-independent effects of AICAr on metabolism, hypoxia, exercise, nucleotide synthesis and cancer. They called for caution in interpretation of AICAr-based studies in the context of understanding AMPK signaling pathway
Comments
- In lines 57/58, it probably helpful to say “..and increase glycogenolysis” as it is the activation of an enzyme.
- In the manuscript, AICAr or acadesine was used in most time concurrently. However, in certain cases, only one of them was used. Please use one way consistently so that your audiences know that you were talking about the same thing, not different ones.
- IN line 171, it should be “..a mouse model…”.
- AMPK has been shown to affect the protein synthesis. Please elaborate on this line of research as well.
Author Response
Thank you very much for your comments on our manuscript. We have addressed all comments and made every attempt to incorporate your suggestions as thorough as possible. All revisions are clearly highlighted using the "Track Changes" function.
The answers to the comments are as follows:
Reviewer 3: The manuscript (Manuscript ID: cells-1164864) entitled “AICAr, a widely used AMPK activator with important AMPK-independent” was by Dora Višnjić, et al. The authors reviewed the present knowledge on AMPK-dependent and AMPK-independent effects of AICAr on metabolism, hypoxia, exercise, nucleotide synthesis and cancer. They called for caution in interpretation of AICAr-based studies in the context of understanding AMPK signaling pathway
Comments
In lines 57/58, it probably helpful to say “..and increase glycogenolysis” as it is the activation of an enzyme.
We greatly appreciate this comment and apologize for the mistake: “…and increase…“ is now added.
In the manuscript, AICAr or acadesine was used in most time concurrently. However, in certain cases, only one of them was used. Please use one way consistently so that your audiences know that you were talking about the same thing, not different ones.
As suggested, acadesine is replaced with AICAr throughout the text.
In line 171, it should be “...a mouse model…”.
We apologize for the mistake; “…a mouse model…“ is introduced.
AMPK has been shown to affect the protein synthesis. Please elaborate on this line of research as well.
As suggested, the effects of AICAr on protein synthesis are elaborated now in a new section of paragraph 1, subheading “AICAr, AMPK, proliferation and cell cycle“ shown in a new Figure 2 (R).
We appreciate your careful evaluation of our work and hope that this revision meets with your approval.
Sincerely yours,
Dora Visnjic